# Adsorption Equilibrium and Thermodynamics of Tea Theasinensins on HP20—A High-Efficiency Macroporous Adsorption Resin

**DOI:** 10.3390/foods10122971

**Published:** 2021-12-02

**Authors:** Jianyong Zhang, Hongchun Cui, Jinjin Xue, Wei Wang, Weiwei Wang, Ting Le, Lin Chen, Ulrich H. Engelhardt, Heyuan Jiang

**Affiliations:** 1Tea Research Institute, Chinese Academy of Agricultural Sciences, Hangzhou 310008, China; zjy5128@126.com (J.Z.); xuejinjin911@tricaas.com (J.X.); zxx531001@126.com (W.W.); lhl5562@126.com (W.W.); letin@tricaas.com (T.L.); 2Tea Research Institute, Hangzhou Academy of Agricultural Sciences, Hangzhou 310024, China; chc1134@126.com; 3Institute of Tea Science, Zhejiang University, Hangzhou 310058, China; chl911003@tricaas.com; 4Institute of Food Chemistry, Technische Universität Braunschweig, 38106 Braunschweig, Germany; u.engelhardt@tu-braunschweig.de

**Keywords:** theasinensins, adsorption, equilibrium, thermodynamics, resin

## Abstract

The separation and preparation of theasinensins have been hot spots in the field of tea chemistry in recent years. However, information about the mechanism of efficient adsorption of tea theasinensins by resin has been limited. In this study, the adsorption equilibrium and thermodynamics of tea theasinensins by a high-efficiency macroporous adsorption HP20 resin were evaluated. The adsorption of theasinensin A, theasinensin B, and theasinensin C on HP20 resin were spontaneous physical reaction processes. Adsorption processes were exothermic processes, and lowering the temperature was beneficial to the adsorption. The Freundlich model was more suitable to describe the adsorption of tea theasinensins. The adsorption equilibrium constant and maximum adsorption capacity of theasinensin A were significantly higher than theasinensin B and theasinensin C, which indicated that the adsorption affinity of theasinensin A was stronger than that of theasinensin B and theasinensin C. The phenolic hydroxyl groups and intramolecular hydrogen bonds of theasinensin A were more than those of theasinensin B and theasinensin C, which might be the key to the resin’s higher adsorption capacity for theasinensin A. The HP20 resin was very suitable for efficient adsorption of theasinensin A.

## 1. Introduction

Theasinensins are important functional components in all six kinds of tea, especially in black tea, green tea, and oolong tea [1,2]. Theasinensins have an important impact not only on the flavor quality of tea, but also on its good biological activity. Theasinensins are important flavor substances in tea soup, which are closely related to the astringency of tea drinks and tea foods [3]. As a food additive, theasinensins could be widely used in non-thermal processing and thermal processing of foods [4]. A variety of biological activities of theasinensins were higher than in some kinds of tea catechins and theaflavins, such as antioxidant [5,6,7], anticancer [8,9], anti-inflammatory [10,11,12,13,14], lipid-lowering and weight reducing [15,16,17,18], cholesterol lowering [19], hypoglycemic [20], and so on. Eight theasinensins have been isolated and identified, including theasinensin A, theasinensin B, theasinensin C, theasinensin D, theasinensin E, theasinensin F, theasinensin G, and theasinensin H [21,22,23,24,25]. The contents of theasinensin A, theasinensin B, and theasinensin C accounted for more than 90% of these eight theasinensins. The separation and preparation of theasinensins were helpful to the study of flavor quality characteristics and biological activity research. The separation and preparation of theasinensins have been hot spots in the field of tea chemistry in recent years.

Theasinensin A and theasinensin B can be separated by HP20 resin column chromatography and Sephadex LH20 gel column chromatography [26]. Theasinensin C and theasinensin D can be separated by HP20 resin, Sephadex LH20 gel, and TSK gel [2,24,25]. Macroporous adsorption resins are an organic polymer adsorbent developed in the 1960s that have been widely used in environmental protection, food, medicine, and other fields. A multicomponent Langmuir isotherm model can be used to describe the adsorptions, and parameters were regressed with high accuracy [27,28]. The adsorptive interaction on resin correlated well with the experimentally measured adsorption affinity and enthalpy [29,30]. However, knowledge and understanding of adsorption equilibrium and thermodynamics of theasinensins on HP20 resin are still scarce.

We studied the adsorption effect of HP20 resin on theasinensins and found that the adsorption selectivity and adsorption capacity of HP20 resin on theasinensins were relatively high, but the selective adsorption equilibrium and thermodynamics were still unclear. We hope to clarify the differences of thermodynamic and kinetic adsorption characteristics of HP20 resin for different kinds of theasinensins. In this study, theasinensin A, theasinensin B, and theasinensin C were the targets, and HP20 macroporous adsorption resin was used as the adsorption material. The adsorption properties of HP20 macroporous adsorption resin at different temperatures and times were compared. The thermodynamic adsorption mechanism was analyzed by adsorption modeling.

## 2. Materials and Methods

### 2.1. Materials and Reagents

Theasinensin A, theasinensin B, and theasinensin C (Figure 1) were prepared by our laboratory. Acetonitrile and methanol were purchased from Merck KGaA (Merck, Darmstadt, Germany). Copper chloride, ascorbic acid, methanol, and phosphoric acid were purchased from Sigma Aldrich (St. Louis, MO, USA). Deionized water (DW) was purified by a Milli-Q system (Millipore, Molsheim, France). 

### 2.2. Preparation of Theasinensins Mixture 

Theasinensins mixture was prepared according to a previously published method [6], with minor modifications. First, 2 mmol of 98% EGCG and 2.8 mmol of 70% EGC were added to 200 mL 26% methanol solution. After EGCG and EGC were completely dissolved, 2 mmol copper chloride was added. The solution was stirred by magnetic stirring at 15 ℃ for 26 h, and 10 mmol ascorbic acid was added. Then, the solution was heated for 15 min at 85 ℃. After cooling, the mixture was concentrated to evaporate MeOH, and the resulting aqueous solution was freeze-dried. Then, the theasinensins mixture was obtained and analyzed by high-performance liquid chromatography (HPLC).

### 2.3. Adsorption Isothermal Experiment

The 1000 mL theasinensins mixture solutions were added to 2 g HP20 resin, respectively. The samples were put into a constant-temperature water bath shaking table at 295 K, 305 K, and 315 K, respectively. The solutions were shaken for 160 min at 100 rpm. The adsorption capacity of HP20 macroporous resin on theasinensins was determined by HPLC. The theasinensin A, theasinensin B, and theasinensin C monomers with purity more than 99% were used as standard materials to calculate the content of theasinensins before and after adsorption. 

The adsorption capacity of theasinensins was calculated according to Equation (1):(1)$$Q_{e}=\frac{m\;(C_{0}-C_{e})}{V}$$
where *C_0_* and *C_e_* are the initial concentration and adsorption equilibrium concentration of theasinensins (mg/L); *Q_e_* is the equilibrium adsorption dose (mg/g); *V* is the volume of methanol solution (mL); and *m* is the mass of adsorbent (g).

According to the experimental data, the isotherm adsorption curves of HP20 macro-porous resin for different theasinensins at different temperatures were drawn. All experiments were performed in triplicate, and average data obtained from the binding studies was used for the calculations of the binding parameters.

### 2.4. Adsorption Thermodynamic Model

The Langmuir, Freundlich, and Sips equations were applied to model the measured adsorption isotherms of theasinensins on HP20 resin for the single system due to their adjustable parameters and simple math expressions.

The adsorption process of theasinensins were fitted by Langmuir isotherm equation (Equation (2)):(2)$$\frac{C_{e}}{Q_{e}}=\frac{1}{Q_{m}K_{L}}+\frac{C_{e}}{Q_{m}}$$
where *Q_e_* is the equilibrium adsorption capacity (mg/g) of the adsorbate on the adsorbent; *C_e_* is the concentration of the adsorbate in the solution at equilibrium (mg/L); maximum adsorption capacity (*Q_m_*) is the saturated adsorption capacity (mg/g); and adsorption equilibrium constant (*K_L_*) is the energy constant.

The adsorption process of theasinensins were fitted by Freundlich isotherm equation (Equation (3)):(3)$$lnQ_{e}=\frac{1}{n}lnC_{e}+{lnΚ}_{L}$$
where $\frac{1}{n}$ is a measure of adsorption strength. When $\frac{1}{n}$ is between 0.1 and 0.5, adsorption is easy, and when $\frac{1}{n}$ ≥ 2, adsorption is difficult. In other words, the higher the n value, the easier the adsorption. *Q_e_* is the equilibrium adsorption capacity (mg/g) of the adsorbate on the adsorbent; *C_e_* is the concentration of the adsorbate in the solution at equilibrium (mg/L); and *K_L_* is the energy constant.

### 2.5. Thermodynamics of Adsorption 

The thermodynamic parameters, the standard Gibbs free energy change (Δ*G*), and the enthalpy change (Δ*H*) could be computed according to thermodynamic laws through the following equations [31,32].
(4)$$\Delta G=\frac{\Omega }{q}=\frac{RT\int_{0}^{C}qdlnC}{q}$$
(5)$$\Delta H=\frac{\left[\frac{ә\left(\Omega /T\right)}{ә\left(1/T\right)}\right]}{q}C$$

Each thermodynamic parameter had its own physical sense in the adsorption system. Δ*G* could be regard as the minimum isothermal work required to load a certain amount of adsorbate on the adsorbent surface. Δ*H* showed the heat effect of the adsorption system [33].

### 2.6. HPLC Measurements of Theasinensins

Analytical reverse-phase HPLC (Shimadzu, LC-20A, Kyoto, Japan) was performed on a 5 μm Cosmosil 5C18-AR II column (Nacalai Tesque Inc., Kyoto, Japan; 4.6 mm i.d. × 250 mm). Mobile phase A was 50 mmol/L phosphoric acid, and mobile phase B was 100% acetonitrile. The cell temperature was maintained at 20 ℃, and the column temperature was maintained at 35 ℃. The gradient for solvent B was set as follows: 0–39 min, 4–30% B; 39–54 min, 30–75% B; 54–60 min, 75–4% B. The flow rate was 0.8 mL/min. The monitoring UV wavelength was set at 280 nm [22]. 

## 3. Results and Discussion

### 3.1. Static Adsorption Isotherm of Theasinensins

The curves of the adsorption equilibrium isothermal conditions of theasinensin A, theasinensin B, and theasinensin C under different temperature conditions were plotted (Figure 2). The adsorption capacity of the column chromatography packing HP20 resin for theasinensin A, theasinensin B, and theasinensin C all decreased with the increase of temperature, which might indicate that the adsorption process of column chromatography packing with HP20 resin to theasinensin A, theasinensin B, and theasinensin C were exothermic. After the adsorption resin surface adsorbs the target substance, the target substance concentration at the adsorption resin interface could be higher than the target substance concentration in the solvent, which would result in a decrease of heat release and free energy in the system. 

The Langmuir adsorption isotherm equation and Freundlich adsorption isotherm equation have been used to analyze the adsorption characteristics of theasinensins by HP20 resin as a column chromatography filler. Fitting data of the equilibrium adsorption capacity of theasinensin A, theasinensin B, and theasinensin C on the HP20 resin are shown in the following Table 1 and Table 2. When the temperature was between 295 K and 315 K, the *K_L_* and *Q_m_* of theasinensin A, theasinensin B, and theasinensin C all decreased with the increase of temperature. The adsorption of theasinensin B and theasinensin C gradually weakened with the increase of temperature. The adsorption process of theasinensin by the column chromatography packing with HP20 resin should not be carried out at high temperature.

The Langmuir adsorption isotherm equation was an ideal model of the surface adsorption for column chromatography packing with HP20 resin. However, in the actual separation and purification process, only a narrow range of adsorption could be recognized on a uniform surface, which indicated that only a narrow range of this equation was applicable [26,27]. Most of it was not applicable. The Freundlich adsorption isotherm equation was used to fit the concentration of the adsorbate in the solution at equilibrium and the equilibrium adsorption capacity of the adsorbate on the adsorbent. Comparing the correlation coefficients of the Langmuir adsorption isotherm equation and Freundlich adsorption isotherm equation of theasinensins, it could be found that the correlation coefficient of the Freundlich adsorption equation was greater than 0.99, which was higher than the correlation coefficient of the Langmuir adsorption isotherm equation, which indicated that the Freundlich adsorption equation was more suitable for evaluating the adsorption of the theasinensin A, theasinensin B, and theasinensin C in the HP20 resin. 

### 3.2. Equivalent Differential Heat of Adsorption of Theasinensins

From the data in Figure 3, it can be seen that the adsorption of theasinensin A, theasinensin B, and theasinensin C by column chromatography packing HP20 resin gradually weakened with the increase of temperature. The adsorption capacity of the HP20 resin for the theasinensins in descending order were theasinensin A, theasinensin B, and theasinensin C. Both *K_L_* and *Q_m_* were significantly higher than pyrogallol-type catechins. This meant that the adsorption driving forces of the aforementioned column chromatography filler HP20 resin to the theasinensins were, in order of magnitude, theasinensin A, theasinensin B, and theasinensin C. The molecular structure of theasinensin A contains relatively more phenolic hydroxyl groups than theasinensin B and theasinensin C. Therefore, theasinensin A was more easily adsorbed on the surface of HP20 resin than theasinensin B or theasinensin C. The hydroxyl structure of the theasinensins formed hydrogen bonds, which produced strong adsorption driving force. The main adsorption force between the HP20 resin and theasinensins might be hydrogen bonds. 

### 3.3. Isometric Differential Adsorption Heat of Theasinensins

The adsorption efficiency was the result of the combined action of various forces between the adsorbate and the adsorbent. Different forces generated different heat in the adsorption process. According to the foregoing analysis, the adsorption process of theasinensin A, theasinensin B, and theasinensin C by column chromatography packing with HP20 resin as an exothermic process. In order to quantitatively describe the heat emission, the Clausius–Clapeyron equation was used to calculate the equivalent differential adsorption heat of each adsorption process. 

There was a good linear relationship between Ln*C* and 1/*T*, which indicated that the adsorption process of theasinensin A, theasinensin B, and theasinensin C obeyed the Clausius–Clapeyron equation. The linear regression method could be used to calculate the corresponding slope of the adsorption amount of theasinensins and the effect of HP20 resin on theasinensin A, theasinensin B, and theasinensin C.

The free energy of adsorption and adsorption entropy of adsorption of theasinensins by HP20 resin have been calculated under different temperature conditions (Table 3). The free energy of adsorption changed to Δ*G*. The adsorption process of the HP20 resin to theasinensin A, theasinensin B, and theasinensin C belonged to the spontaneous physical reaction process. The theasinensins’ adsorption free energy Δ*G* were, in order, theasinensin A, theasinensin B, and theasinensin C. The adsorption enthalpy changes of HP20 resin for theasinensin A, theasinensin B, and theasinensin C were all negative, which indicated that the adsorption process was exothermic. 

The adsorption and separation function of column chromatography mainly relied on intermolecular force, hydrogen bond, coordination bond, hydrophobic interaction, ionic bond, and other interactions to achieve its adsorption and separation function. Kristl et al. tested the range of adsorption heat caused by different types of forces [27], and found that the adsorption heat caused by van der Waals force was 4–10 KJ/mol, that caused by water-dispersing bond force was about 5 KJ/mol, that caused by hydrogen bond force was 2–40 KJ/mol, that caused by coordination bond exchange was greater than 40 KJ/mol, and that caused by chemical bond force was greater than 60 KJ/mol. The adsorption thermodynamic parameters of the HP20 resin for theasinensin A, theasinensin B, and theasinensin C were 10–40 KJ/mol, which belong to hydrogen bond force adsorption.

The HP20 resin had weak acidity with theasinensin A, theasinensin B, and theasinensin C, and would not form ionic bonds, covalent bonds, or coordination bonds. The main force could only be hydrogen bonding and hydrophobic interaction. These hydrogen bonding effects included the hydrogen bonding between HP20 resin and the hydroxyl group of theasinensins, the hydrogen bonding between the hydroxyl group of theasinensins and water molecules, and the hydrogen bonding between HP20 resin and water molecules.

As a result, the adsorption capacity of HP20 to theasinensin A, theasinensin B, and theasinensin C increased with the increase of equilibrium concentration (Figure 4). However, as the adsorption capacity of theasinensins by HP20 resin increased, the equivalent differential adsorption heat value decreased instead. For adsorption sites with uniform adsorption capacity, there was no force between adsorbate molecules. The heat of adsorption had nothing to do with the amount of adsorption. However, in the case of uneven surface capacity of the adsorbent, it was necessary to consider the interaction between theasinensins molecules. In addition, the adsorption heat changed with the surface coverage θ of the HP20 resin.

There were many high-capacity adsorption centers on the surface of the HP20 resin. The adsorbent was first adsorbed on the part with large surface energy. The adsorption heat was large, but the activation energy was weak. It then gradually transferred to the part with low surface energy. At the same time, the heat released decreased accordingly. The theasinensins were firstly adsorbed on the parts with high differential adsorption heat, and then gradually transferred to the parts with low differential adsorption as the coverage increased. In addition, with the continuous increase of the surface coverage θ of theasinensins on the aforementioned column chromatography filler of HP20 resin, the mutual repulsion force between adsorbate molecules became stronger.

### 3.4. Comparison of Adsorption Capacity of Theasinensins by HP20 Resin

The adsorption isotherm of HP20 resin to theasinensins was studied (Figure 4 and Figure 5). When the equilibrium concentration was the same, the adsorption capacity of theasinensin A by HP20 resin was greater than that of theasinensins B, and theasinensin B by HP20 resin was greater than that of theasinensins C. The main reason was that the polyphenolic hydroxyl group of theasinensin A was better than theasinensin B and theasinensin C. The extra phenolic hydroxyl group of theasinensin A could bond with the hydroxyl group of the HP20 resin, which increased the adsorption driving force, making the adsorption capacity larger.

When the equilibrium concentration was the same, the adsorption capacity of theasinensin C by HP20 resin was lower than that of theasinensin B or theasinensin A. The main reason was that theasinensin A had intramolecular hydrogen bonds, which reduced the ability of HP20 resin to form hydrogen bonds with it. This also proved that the column chromatography packing with HP20 resin adsorbed theasinensins through hydrogen bonding force.

## 4. Conclusions

The selective separation mechanism based on adsorption thermodynamic analysis was explored. The adsorption process of theasinensins by column chromatography packing with HP20 resin conforms to the Freundlich adsorption isotherm equation (R^2^ value is 0.9968–0.9995). When the temperature was between 295 K and 315 K, the adsorption equilibrium constant (*K_L_*) and the maximum adsorption capacity (*Q_m_*) of tea theasinensins decreased with the increase of temperature. The results showed that the adsorption process of HP20 on tea theasinensins should not be carried out at relatively high temperature. When the temperature was the same, the *K_L_* and *Q_m_* of theasinensin A were significantly higher than theasinensin B and theasinensin C. This showed that the filler had a stronger adsorption affinity for theasinensin A than theasinensin B and theasinensin C. The adsorption process of the filler on theasinensin A, theasinensin B, and theasinensin C was a spontaneous physical reaction process. The higher the temperature, the stronger the driving force of the spontaneous reaction. The adsorption processes were exothermic processes. Low temperature was conducive to the progress of the reaction. The entropy changes in the adsorption process were all negative values. The phenolic hydroxyl groups and intramolecular hydrogen bonds might be the key to the resin’s higher adsorption capacity for theasinensin A. The HP20 was an efficient resin for adsorbing theasinensins. This article focused on the adsorption equilibrium and thermodynamics of theasinensin A, B, and C in the theasinensins mixture, and the possible mechanism of high adsorption characteristics. In future work, the recovery characteristics, yield, and rate of theasinensins will be optimized in order to have an extract suitable for food applications. These research contents could provide a theoretical basis for their application in the flavoring of food and functional food.

## Figures and Tables

**Figure 1 foods-10-02971-f001:**
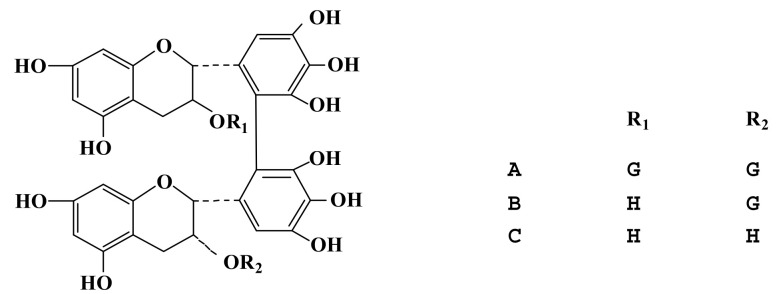
Structural formula of theasinensin A, theasinensin B, and theasinensin C.

**Figure 2 foods-10-02971-f002:**
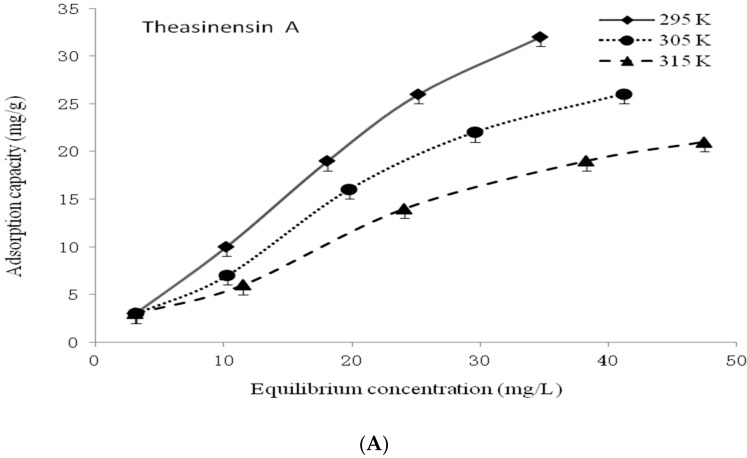
Adsorption equilibrium isotherms of theasinensin (**A**–**C**) on column chromatography packing with HP20 resin.

**Figure 3 foods-10-02971-f003:**
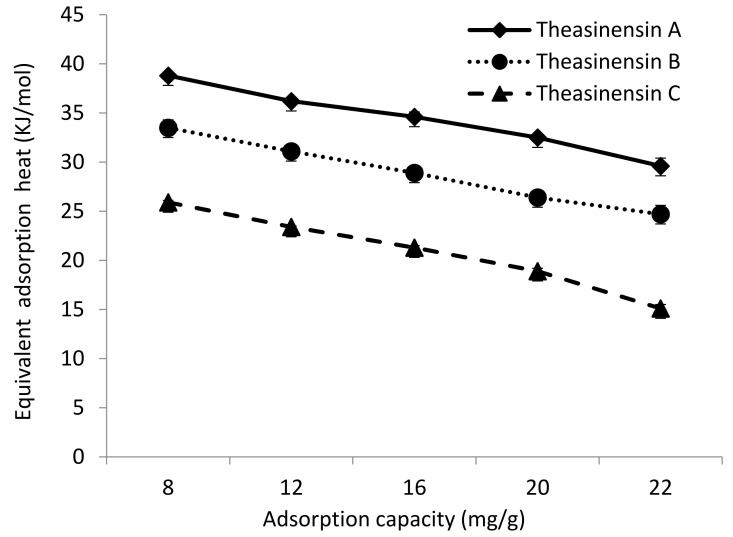
Theasinensins with equivalent adsorption heat under different adsorption capacities.

**Figure 4 foods-10-02971-f004:**
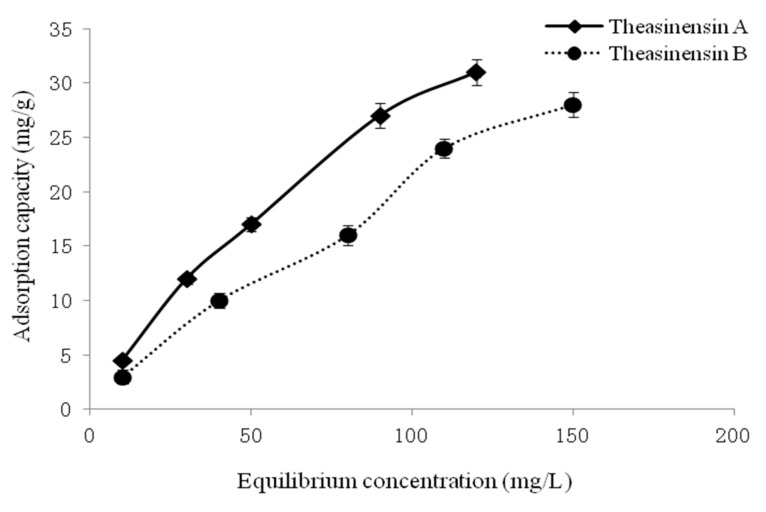
Adsorption isotherms of theasinensins by HP20 resin.

**Figure 5 foods-10-02971-f005:**
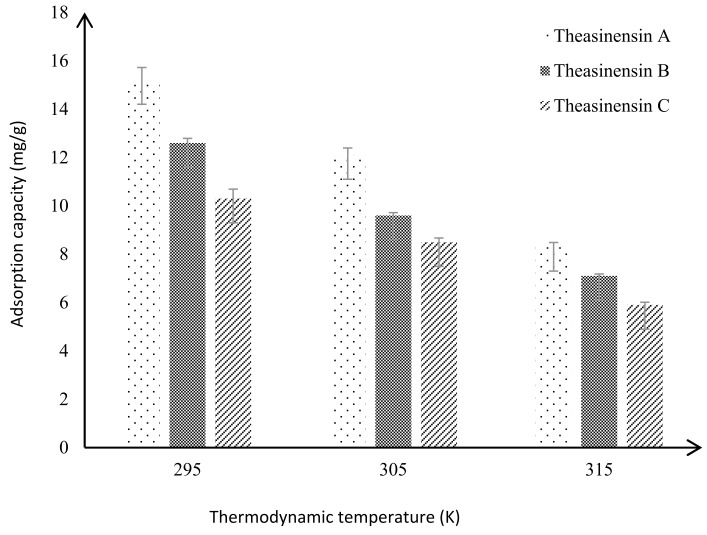
Comparison of adsorption capacity of theasinensins under different thermodynamic temperatures.

**Table 1 foods-10-02971-t001:** Adsorption parameters of the Langmuir adsorption equation of theasinensins.

Adsorbate	Thermodynamic Temperature (K)	Regression Equation	*Q_m_* (mg/g)	*K_L_* (mg/L)	Correlation Coefficient (*R^2^*)
Theasinensin C	295	y = 0.017x + 1.225	40.53 ± 1.55	0.016 ± 0.0006	0.9623
	305	y = 0.015x + 2.328	43.22 ± 1.22	0.012 ± 0.0003	0.9712
	315	y = 0.016x + 3.026	38.65 ± 1.06	0.009 ± 0.0001	0.9745
Theasinensin B	295	y = 0.019x + 0.565	39.85 ± 1.12	0.058 ± 0.0011	0.9587
	305	y = 0.017x + 1.126	40.98 ± 1.49	0.033 ± 0.0009	0.9766
	315	y = 0.016x + 1.779	38.88 ± 1.36	0.019 ± 0.0008	0.9682
Theasinensin A	295	y = 0.042x + 0.298	22.93 ± 1.17	0.432 ± 0.0093	0.9721
	305	y = 0.042x + 0.237	24.81 ± 1.14	0.365 ± 0.0088	0.9621
	315	y = 0.042x + 0.514	20.67 ± 1.01	0.253 ± 0.0065	0.9521

**Table 2 foods-10-02971-t002:** Adsorption parameters of the Freundlich adsorption equation of theasinensins.

Adsorbate	Thermodynamic Temperature (K)	Regression Equation	*n*	*K_L_* (mg/L)	Correlation Coefficient (*R^2^*)
Theasinensin C	295	y = 0.5988x + 0.357	1.685	0.016 ± 0.0005	0.9968
	305	y = 0.6233x + 0.225	1.556	0.012 ± 0.0003	0.9979
	315	y = 0.6985x + 0.347	1.438	0.009 ± 0.0001	0.9983
Theasinensin B	295	y = 0.4261x + 1.456	2.690	0.058 ± 0.0016	0.9966
	305	y = 0.5467x + 1.385	2.153	0.033 ± 0.0011	0.9972
	315	y = 0.5568x + 0.958	1.682	0.019 ± 0.0008	0.9979
Theasinensin A	295	y = 0.4668x + 1.699	2.893	0.432 ± 0.012	0.9991
	305	y = 0.4598x + 2.329	2.741	0.365 ± 0.020	0.9981
	315	y = 0.4622x + 2.258	2.855	0.253 ± 0.008	0.9995

**Table 3 foods-10-02971-t003:** Adsorption thermodynamic parameters of HP20 resin separation system.

Adsorbate	*Q* (mg/g)	Δ*H* (KJ/mol)	Δ*G* (KJ/mol)		
			295K	305K	315K
Theasinensin A	8	38.8 ± 1.21	7.17 ± 0.25	7.32 ± 0.28	7.68 ± 0.32
	12	36.2 ± 1.33			
	16	34.6 ± 1.25			
	20	32.5 ± 1.18			
	22	29.6 ± 0.95			
Theasinensin B	8	33.5 ± 1.65	5.31 ± 0.11	5.68 ± 0.16	5.89 ± 0.12
	12	31.1 ± 1.03			
	16	28.9 ± 1.21			
	20	26.4 ± 1.05			
	22	24.7 ± 0.88			
Theasinensin C	8	25.9 ± 0.75	4.36 ± 0.08	4.15 ± 0.06	3.97 ± 0.03
	12	23.4 ± 0.66			
	16	21.3 ± 0.53			
	20	18.9 ± 0.26			
	22	15.1 ± 0.39			

## Data Availability

Not applicable.

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
