# Peer review of "Adsorption Equilibrium and Thermodynamics of Tea Theasinensins on HP20—A High-Efficiency Macroporous Adsorption Resin"

_foods, 2021, doi:10.3390/foods10122971_

Round 1

Reviewer 1 Report

The authors present a study about the adsorption of important tea components using HP20 resin.

The article is well organized and the results are interesting but it seems to be incomplete.

  1. What is the main objective of this study? Just study the adsorption of theasinensis in a new material or present a way to recover/separate this family of compounds from sample matrices?
  2. How can theasinensis be recovered from this resin? What is the recovery yield?
  3. Did the author study each one of the theasinensis compounds isolated or a mixture was also used to perform the adsorption studies? This information is not clear in the material and methods section.
  4. Which are the main applications of these compounds in the food industry? This information should be added to the Introduction.
  5. The authors should check the text for typos and small errors.

Author Response

Response to Reviewer 1 Comments

Point 1: What is the main objective of this study? Just study the adsorption of theasinensis in a new material or present a way to recover/separate this family of compounds from sample matrices?

Response 1:

Thank you very much for your constructive comments. The main objective of this study is explaining the mechanism of efficient adsorption of tea theasinensins by the HP20 resin, a high efficiency microporous adsorption resin. We hope that this study can provide a theoretical basis for the efficient separation of theasinensins.

Point 2: How can theasinensis be recovered from this resin? What is the recovery yield?

Response 2: 

These are very good questions. This article focused on the  adsorption equilibrium and thermodynamics of theasinensin A, B, C in the theasinensins mixture. Limited to the length requirements of the journal, the results of the recovery characteristics and recovery rate of theasinensins are not reflected in this paper. What we can share are that the theasinensins could be recovered by 80% methanol solution from this resin, and the recovery yield could reach to  95.35%.

Point 3: Did the author study each one of the theasinensis compounds isolated or a mixture was also used to perform the adsorption studies? This information is not clear in the material and methods section.

Response 3: 

We were sorry about that we did not clearly explaining the main objects and purposes of theasinensins adsorption properties research. We have studied the theasinensins mixture to perform the adsorption studies. We focused on the  adsorption equilibrium and thermodynamics of theasinensin A, B, C in the theasinensins mixture. This information can be obtained from lines 87 to 92 of the revised manuscript, which described like this “Then the theasinensins mixture was obtained and analyzed by HPLC”, “The 10 mg•ml-1 theasinensins solutions were added to 2 g HP20 respectively. The samples were put into constant temperature water bath shaking table at 295 K, 305 K and 315 K respectively” .

Point 4: Which are the main applications of these compounds in the food industry? This information should be added to the Introduction.

Response 4: 

Thank you very much for your constructive comments. The theasinensins are important flavour substances in tea soup, which is closely related to the astringency of tea drinks and tea foods. As a food additive, theasinensins can be widely used in non-thermal processing and thermal processing foods. These have been added to the introduction, which were added to lines 33 through 35 of this article.

Point 5: The authors should check the text for typos and small errors.

Response 5: 

Thank you very much for your good comments. We have carefully checked the text for typos and small errors, which can be found in section of author's unit, abstract,  introduction, materials and methods, results and discussion, concluting remarks, such as lines 7-13, 16, 27 28, 35-38, 41, 44, 45, 48, 50, etc in the revised manuscript.

Reviewer 2 Report

The present manuscript studied the adsorption equilibrium and thermodynamics of tea theasinensins  by  a  high  efficiency  macroporous  adsorption  HP20  resin. The adsorption process of theasinensins by column chromatography packing with HP20 resin conformed to the Freundlich adsorption isotherm equation; further the results showed that the adsorption process of HP20 on tea theasinensins should not be carried out at relatively high temperature.

The work is interesting and addresses the scarce information about the mechanism of efficient adsorption of tea theasinensins by resin, focusing on a “hot spots” in the field of tea chemistry in recent years.

  • My major concern is about the HPLC measurements of theasinensins. The section 2.6 is maigre and lacks important details. The overall instrument is not described. The particle size of the C18 column is not reported. It is weird to use a flow-rate of 350 μL/min with a 4.6. mm I.D. column. Further the authors are strongly encouraged to provide an HPLC chromatogram with proper peak identification with a new section in Results and Discussion concerning the HPLC results.
  • English needs to be significantly polished. Use the past tense throughout the whole manuscript. Also the impersonal form.

Author Response

Response to Reviewer 2 Comments

Point 1: My major concern is about the HPLC measurements of theasinensins. The section 2.6 is maigre and lacks important details. The overall instrument is not described. The particle size of the C18 column is not reported. It is weird to use a flow-rate of 350 μL/min with a 4.6. mm I.D. column. Further the authors are strongly encouraged to provide an HPLC chromatogram with proper peak identification with a new section in Results and Discussion concerning the HPLC results.

Response 1: 

Thank you very much for your constructive comments. We were sorry about that the article lacked some important details and clerical error. The HPLC instrument specific information and particle size of the C18 column have been supplemented and improved into the article. The flow rate was 0.8 ml/min(in the line 137 of the original manuscript) , which could be found in the line 141 of the revised manuscript. We were very sorry about that “The flow rate and injection volume were 350 µL/min and 2 µL/min, respectively“ was a serious clerical error, which should not appear in this article. Limited to the length requirements of the journal, the results of the HPLC chromatogram of theasinensins with proper peak identification was not reflected in this paper.

Point 2: English needs to be significantly polished. Use the past tense throughout the whole manuscript. Also the impersonal form.

Response 2: 

Thank you very much for your constructive comments. Many inappropriate English writing methods in the thesis have been modified, including tense, grammar and so on.  

    In section of author's unit of the revised manuscript,  the specific information of the first and second research institute had been supplemented, which could be found in the lines 7-13 of the revised manuscript.

    In the abstract section of the revised manuscript,  some minor mistakes and writing had been modified and added, which could be found in the lines 16, 27, 28 of the revised manuscript.

    In the introduction section of the revised manuscript,  some tense  and grammar mistakes and writing style had been modified and added, which could be found in the lines 32, 35-38, 41, 44, 45, 48, 50-55, 58, 61-62 of the revised manuscript.

    In the materials and methods section of the revised manuscript, some tense  and grammar mistakes had been modified and added, which could be found in the lines 71, 87, 88, 93, 94, 99-101, 110, 113-117, 120-124, 127, 131-133, 136-137 of the revised manuscript.

    In the results and discussion section of the revised manuscript,  some tense  and grammar mistakes and writing style had been modified and added, which could be found in the lines 148, 150, 152, 154, 155, 162, 164-165, 168, 174-179, 182-185, 189-192, 194-201, 205-206, 209-214, 217-225, 230-231, 236, 238-244, 246-252, 257-265, 268-273, 278-282 of the revised manuscript.

    In the concluting remarks section of the revised manuscript, some tense  and grammar mistakes had been modified and added, which could be found in the lines 288, 290, 292-299 of the revised manuscript.

Round 2

Reviewer 1 Report

The authors answered satisfactorily to all the questions raised by the reviewers. 

The authors should add if possible, a reference to the information added to the introduction, from lines 33 to 35.

Regarding the recovering of theasinensis from the resin, the authors could add to the conclusion something like:

This article focused on the adsorption equilibrium and thermodynamics of theasinensin A, B, C in the theasinensins mixture. In future work, the recovery characteristics, yield and rate of theasinensins will be optimized in order to have an extract suitable for food applications.

Author Response

Point 1: The authors should add if possible, a reference to the information added to the introduction, from lines 33 to 35.

Response 1:

Thank you very much for your constructive comments. The references [3] and [4] had been added to lines 33-35 in the introduction. Accordingly, the order of the whole references had been adjusted again. And, the references cited by the main body of the paper had been marked one by one.

Point 2: Regarding the recovering of theasinensis from the resin, the authors could add to the conclusion something like:This article focused on the adsorption equilibrium and thermodynamics of theasinensin A, B, C in the theasinensins mixture. In future work, the recovery characteristics, yield and rate of theasinensins will be optimized in order to have an extract suitable for food applications.

Response 2: 

These are very good constructive comments. The outlook had been added to the conclusion. The supplementary contents were:

The phenolic hydroxyl groups and intramolecular hydrogen bondsin might be the key to the resin's higher adsorption capacity for theasinensin A. The HP20 was a kind efficient resin for adsorbing theasinensins. This article focused on the adsorption equilibrium and thermodynamics of theasinensin A, B, C in the theasinensins mixture, and possible mechanism of high adsorption characteristics. In future work, the recovery characteristics, yield and rate of theasinensins will be optimized in order to have an extract suitable for food applications. These research contents can provide theoretical basis for its application in flavor food and functional food. 

Reviewer 2 Report

The authors have addressed most ot the Reviewer concerns.

Only a few further modifications:

- Use present tense in lines 33,34,35,42 and 62.

Author Response

Point 1: The authors have addressed most ot the Reviewer concerns.

Only a few further modifications:

- Use present tense in lines 33,34,35,42 and 62.

Response 1: 

Thank you very much for your constructive comments. The present tense had been used in lines 33,34,35,42 and 62.
